# Optimization Conditions to Obtain Cationic Polyacrylamide Emulsion Copolymers with Desired Cationic Degree for Different Wastewater Treatments

**DOI:** 10.3390/polym15122693

**Published:** 2023-06-15

**Authors:** Tung Huy Nguyen, Linh Pham Duy Nguyen, Thao Thi Phuong Nguyen, Minh Xuan Anh Le, Linh Thi Thuy Kieu, Huong Thi To, Thanh Tien Bui

**Affiliations:** 1Center for Polymer Composite and Paper, School of Chemical Engineering, Hanoi University of Science and Technology, Hanoi 11600, Vietnam; thao.nguyenthiphuong@atpcorporation.com.vn; 2Department of Pharmaceutical Chemistry and Pesticides Tech, School of Chemical Engineering, Hanoi University of Science and Technology, Hanoi 11600, Vietnam; lab01@atpcorporation.com.vn; 3Department of Chemical Process Equipment, School of Chemical Engineering, Hanoi University of Science and Technology, Hanoi 11600, Vietnam; linh.kieuthithuy@atpcorporation.com.vn; 4Lab of Petrochemical Refining & Catalysis, School of Chemical Engineering, Hanoi University of Science and Technology, Hanoi 11600, Vietnam; huong.tothi@atpcorporation.com.vn

**Keywords:** cationic polyacrylamide, cationic degree, response surface model, wastewater treatment

## Abstract

The synthesis of cationic polyacrylamides (CPAMs) with the desired cationic degree and molecular weight is essential for various industries, including wastewater treatment, mining, paper, cosmetic chemistry, and others. Previous studies have already demonstrated methods to optimize synthesis conditions to obtain high-molecular-weight CPAM emulsions and the effects of cationic degrees on flocculation processes. However, the optimization of input parameters to obtain CPAMs with the desired cationic degrees has not been discussed. Traditional optimization methods are time-consuming and costly when it comes to on-site CPAM production because the input parameters of CPAM synthesis are optimized using single-factor experiments. In this study, we utilized the response surface methodology to optimize the synthesis conditions, specifically the monomer concentration, the content of the cationic monomer, and the content of the initiator, to obtain CPAMs with the desired cationic degrees. This approach overcomes the drawbacks of traditional optimization methods. We successfully synthesized three CPAM emulsions with a wide range of cationic degrees: low (21.85%), medium (40.25%), and high (71.17%) levels of cationic degree. The optimized conditions for these CPAMs were as follows: monomer concentration of 25%, content of monomer cation of 22.5%, 44.41%, and 77.61%, respectively, and initiator content of 0.475%, 0.48%, and 0.59%, respectively. The developed models can be utilized to quickly optimize conditions for synthesizing CPAM emulsions with different cationic degrees to meet the demands of wastewater treatment applications. The synthesized CPAM products performed effectively in wastewater treatment, with the treated wastewater meeting the technical regulation parameters. ^1^H-NMR, FTIR, SEM, BET, dynamic light scattering, and gel permeation chromatography were employed to confirm the structure and surface of the polymers.

## 1. Introduction

Industrial wastewater comprises various components, including suspended solids, dissolved ions, and organic and inorganic particles, among others. Wastewater treatment plays an important role in sustainable development and environmental safety. Flocculation is an essential industrial process for wastewater treatment [1]. One of the most used flocculants for wastewater treatment and sludge de-watering is a water-soluble polymer produced by polymerizing acrylamide monomers or co-polymerizing acrylamide monomers with other monomers. Cationic polyacrylamides (CPAMs) are widely utilized due to their excellent performance in flocculation and sludge dewatering [2]. Numerous studies have been conducted on CPAM synthesis technologies, including grafting, free radical polymerization, and polymer modification [3]. The grafting approach in CPAM synthesis results in a biodegradable but unstable product with a short shelf life [3]. A more efficient method for producing CPAMs with a high molecular weight, faster reaction rate, higher conversion efficiency, and simpler temperature control is through free radical polymerization by inverse emulsion [3,4,5,6,7].

Previous research, such as the studies by Ma et al., successfully synthesized CPAMs with cationic degrees of up to 40 percent for the treatment of wastewater with high kaolin content [8,9]. Fu et al. optimized CPAM dosage, stirring time, and wastewater pH for flocculation of kaolin-based wastewater using the response surface methodology [10]. Zhou et al. investigated the effect of molecular weight and cationic degree of commercial CPAM products on sludge dewatering and moisture evaporation [11]. Sun et al. synthesized a new CPAM (P(DAC-MAPTAC-AM)) using UV-induced polymerization technology and tested the flocculation performance of CPAMs with different cationic degrees (20, 30, and 40%) [12]. Wang et al. applied CPAMs in algae removal from shipboard rotary drum filters and found that CPAMs with a 30% cationic degree exhibited excellent flocculation effectiveness [13]. The cationic degree of CPAMs is an important parameter relevant to their application in wastewater treatment [14,15,16]. Different cationic degrees result in different levels of efficiency in wastewater treatment processes, such as sewage sludge dewatering [11]. Depending on the market demands, a CPAM product with a specific cationic degree may be required for a particular application. For example, biosolids dewatering flocculants typically require a cationic degree of 40–50%. To dewater very young, high f/m, pure bio-sludges, CPAMs with a cationic degree of 60–70% would be needed. Paper mill sludge dewatering flocculants would require CPAMs with a cationic degree of 10–30%. Due to the short storage time of CPAMs, on-site mass production is preferable to importing them from overseas. Recent studies have focused on the synthesis of CPAM products for specific purposes. They have investigated the effects of the cationic degree of CPAMs on flocculation performance, but none of them have shown the optimized conditions for quickly obtaining CPAMs with specific cationic degrees resulting in longer product development time, higher production costs for CPAM, and an inability to meet market demand.

Therefore, establishing a relationship between the cationic degree of CPAMs and input parameters such as monomer concentration, content of the cationic monomer, and initiator content is necessary. In this study, we present a method for quickly optimizing the conditions of CPAM emulsion copolymerization to produce CPAMs with different cationic degrees for various wastewater treatment applications based on user demands. This study helps reduce the research time for new products, rapidly satisfy market demand, enable local production of CPAM emulsions, minimize the product’s expiry date due to long shipping processes, and reduce the production costs of CPAMs. 

The structures of the synthesized polymers were verified using ^1^H-NMR and FTIR spectroscopy. The surface microstructures of the polymers were observed using scanning electron microscopy (SEM), and the surface area was determined through Brunauer–Emmett–Teller (BET) analysis. The molecular weight and distribution of the CPAM were confirmed using a viscosity meter and gel permeation chromatography (GPC). The particle size distribution of the CPAM was determined using the dynamic light scattering (DLS) method. 

## 2. Materials and Methods

### 2.1. Chemicals

A cation monomer solution 74% in *N,N,N* -trimethyl-2-[(2-methyl-1-oxo-2-propenyl)oxy]-chloride (DMC) was purchased from Guangchuangjing Company (Shanghai, China) as an aqueous solution. Industrial monomer acrylamide (AM, 97%) was purchased from Lanjie Tap Water Company (Chongqing, China). Iso par L (Exxon), Span 80 (99%), and Tween 85 (99%) were purchased from Beijing Chemical Reagent Company (Beijing, China). 2,2′-azobis(2-methylpropionamidine) dihydrochloride (V50, 99%), potassium persulphate (K_2_S_2_O_8_, 99%), and sodium bisulfite (NaHSO_3_, 90%) were purchased from Tokyo Chemical Industry Co., Ltd. Isopropanol (99%), iodine (99%), ethanol (99.9%), potassium iodide (99%), mercury chloride (99%,) sodium thiosulfate (99%), starch soluble (99%), silver nitrate (99%), and nonylphenol ethoxylate (NP-9) were supplied by Xilong Scientific Co., Ltd. (Guangdong, China).

Chemicals that have been used to treat wastewater, including poly aluminum chloride (PAC, 31%), were purchased from Hengyang Jianheng Industry Development Co., Ltd. (Hengyang, Hunan, China). Hydrochloric acid solution (HCl, 45%), sulfuric acid solution (H_2_SO_4_, 50%), and sodium hydroxide solution (NaOH, 32%) were purchased from Truong An Technology and Trading Co., Ltd. (Ho Chi Minh, Vietnam). Calcium chloride (CaCl_2_, 99%) was purchased from Weifang Qingtong Chemical Co., Ltd. (Shandong, China). 

### 2.2. Preparation of CPAM 

The CPAM was prepared using the reverse emulsion copolymerization method (Figure 1). A reaction tank with a mechanical stirrer, a thermometer for automatic control temperature, a condenser, and a system for high-purity nitrogen were used for each reaction (Figure 2). Throughout the reaction, the nitrogen gas was continually aerated. Phase 1 of the reaction was initiated using UV light and a photo-initiator. Phase 2 of the reaction, which was completed with stirring and the gradual addition of a redox initiator system, was when the factors influencing the polymerization reaction—monomer concentration and the proportion of the DMC monomer—were examined. The synthesis reaction was followed by the formation of an inverse emulsion cation polymer.

### 2.3. Determination of Molecular Weight and Conversion of CPAM

Utilizing NP-9, the W/O (water in oil) emulsion product was changed into an O/W emulsion. After that, isopropyl alcohol was gradually added until the O/W emulsions were transparent in a 100 mL beaker. In a vacuum oven cabinet, the precipitate was filtered and then dried at 45 °C to obtain a constant mass, and “filtrate X” was utilized for titration and conversion. 

Using the HIP approach, the residual content of monomers (AM and DMC) was used to calculate the overall monomer conversion. The excess AM and DMC were added after introducing the HIP solution (i.e., I_2_ and HgCl_2_ in ethanol). I_2_ then interacted with HgCl_2_ to generate ICl, which was then added. To obtain I_2_, the extra ICl was reduced with KI. I_2_ was titrated with a Na_2_S_2_O_3_ solution to determine the total monomer conversion.

The conversion was calculated using Equation (1):(1)$$H\left(\%\right)=\left(C-1/2\times \left(\left(V_{0}-V\right)\times N\right)/V_{i}\right)/C$$
where *C* is the volume concentration of the initial monomer; *V*_0_ and *V* (mL) are the volume of Na_2_S_2_O_3_ used to titrate the residual monomers in the blank sample and in the sample at time *i*; *N* is the concentration of the Na_2_S_2_O_3_ solution; and *V_i_* (mL) is the volume of the reaction mixture at the time *i*. The conversion measurements were carried out in triplicate. 

Approximately 200 g of deionized water was used to dry the dissolved CPAM. The viscosity of the polymers was measured using an Ubbelohde viscometer. The Mark–Houwink–Sakurada equation was utilized to determine the polymers’ molecular weight:*η* = *K* × *M*^*α*^(2)
where *η* and *M* are the viscosity and molecular weight of the polymer; and *K* and *α* are the characteristic constants of the polymers and solvents [17]. The molecular weights of the CPAM and its distribution were confirmed by gel permeation chromatography.

### 2.4. Determination of the Cationic Degree of CPAM

The cationic degree (*DC*) of the CPAM was determined by titration of a silver nitrate solution. Approximately 0.3 g of the dried CPAM was dissolved in 150 mL distilled water. After 5 h, a few drops of potassium dichromate indicator were added, and then the mixture was titrated using a 0.1 M silver nitrate (AgNO_3_) solution until the solution turned from yellow to brick red. The cationic degree is expressed by the following equation:*DC* (*%*) = (*M* × *N* × (*V* − *V*_0_))/*W*(3)
where *M* is the molecular weight of cationic monomer; *N* is the molar concentration of the silver nitrate solution; *V* and *V*_0_ are the volumes of the silver nitrate solution reacted with the sample and the blank, respectively; and *W* is the amount of the CPAM [18].

### 2.5. Particle Size Distribution of CPAM

The particle size distribution of the polyacrylamide cationic nanoparticles was determined by the dynamic light scattering (DLS) method (instrument: Nanoparticle Analyzer Horiba SZ-100).

### 2.6. Structure Analysis

FT-IR spectra were acquired using an FT-IR spectrometer (model IRAffinity-1S, Shimadzu, Japan). ^1^H-NMR spectra were collected using a Bruker Avance Neo 600 MHz spectrometer.

SEM images of the polymer were observed using a scanning electron microscope (model S4800, Hitachi, Japan) at an accelerating voltage of 5 kV. The surface area and nitrogen adsorption–desorption analysis of CPAM was performed using a MicroActive for TriStar II Plus 2.03 (Micromeritics Instrument Corporation, Norcross, GA, USA).

### 2.7. Statistical Analysis

The experimental data were processed using Design-Expert 11.1 software (Stat-Ease, Minneapolis, MN, USA). The Box–Behnken method [19,20,21] was used to design the experiments.

### 2.8. Wastewater Treatment Testing Using Synthesized CPAM

In this study, the amount of polymer coagulant greatly affected the flocculation process. The wastewater sample was stirred in a magnetic stirrer with pH adjustment chemicals such as Ca(OH)_2_, PAC, and CPAM coagulant. The treated wastewater was then analyzed to obtain the indicators of TSS (total suspended solids), BOD (biological oxygen demand), and COD (chemical oxygen demand). TSS, COD, and BOD were measured using a UV Probe 254+ meter (France). The turbidity was obtained using a turbidity meter (model Thermo Scientific™ Eutech TN-100, Loughborough, UK).

## 3. Results and Discussion

### 3.1. Analysis Structure of Synthesized CPAM

Three CPAM samples were synthesized, CPAM-1, CPAM-2, and CPAM-3, which had cationic degrees of 21.85%, 40.25%, and 71.17%, respectively. The FTIR results of the CPAM and functional groups assignment are shown in Figure 3. The peaks A and E with frequencies of 3340 cm^−1^ and 1658 cm^−1^ correspond to the stretching vibration of two groups (i.e., –NH_2_ and C=O) in the amide groups of the AM monomer [22]. The 2930 cm^−1^ peak (peak C) corresponds to –CH_3_ and –CH_2_– [9]. The adsorption peak at 1460 cm^−1^ (peak F) was from –CH_2_– flexural vibrations in –CH_2_–N^+^ (CH_3_)_3_ [23]. The absorption peaks of 1730 cm^−1^ and 1130 cm^−1^ (peaks D and G) are from the C=O and C–O–C bonds in the ester group (–COOCH_2_–) of the cation monomer DMC, respectively. The 944 cm^−1^ adsorption peak (peak H) was assigned to the quaternary ammonium groups [24]. Peak B at 3170 cm^−1^ corresponds to OH groups [25]. The FTIR results proved that AM and DMC were copolymerized with different cationic degrees. Figure 3 shows that the transmittance intensity of the quaternary ammonium group (peak at 944 cm^−1^) increased with increasing cation content in the CPAM (CPAM-1 < CPAM-2 < CPAM-3).

Figure 4 shows the ^1^H–NMR spectra of the three CPAM samples with different DC values. The chemical shift at 0.92 ppm corresponds to the protons in the –CH_3_ group (peak a). Two peaks at δH = 1.75 ppm and δH = 2.22 ppm correspond to the protons of –CH_2_– (peak b) and –CH– (peak c), respectively. The peak at δH = 3.27 ppm corresponds to the protons of the –N+(CH_3_)_3_ group (peak d). The peak at 4.55 ppm corresponds to the peak e of O=C–O–CH_2_^+^. The sharp peaks at δH = 3.79 ppm correspond to the proton of –N^+^CH_2_– (peak g). Lastly, the chemical shift at 5.35 ppm corresponds to the protons of O=C–NH_2_ (peak f). The ^1^H–NMR spectral results agreed well with previous data that indicated that CPAM copolymers were successfully synthesized from AM and DMC [26]. The chemical shifts of 5.6–6.2 ppm (corresponding to CH_2_=CH– and CH_2_=C–) [27] did not show any peaks in Figure 4, which means that the monomers reacted completely and the reactions had a high conversion rate. The peak intensity of the quaternary ammonium group (peak d) of the cation monomer DMC increased when the content of the DMC monomer in the CPAM copolymer increased (CPAM-1 < CPAM-2 < CPAM-3).

Figure 5 shows the molecular weight distribution of the CPAM samples. CPAM-1 (DC = 21.85%) had a number-average molecular weight (M_n_) of 5,515,200 g/mol and weight-average molecular weight (M_w_) of 18,984,000 g/mol. The polydispersity index D (M_w_/M_n_) was 3.4421. CPAM-2 (DC = 40.25%) had an M_n_ of 7,446,100 g/mol, M_w_ of 19,218,000 g/mol, and D = 2.581. CPAM-3 (DC = 71.17%) had an M_n_ of 9,633,400 g/mol, M_w_ of 20,425,000 g/mol, and D = 2.1202. The CPAM samples synthesized by reverse emulsion copolymerization had molecular weight distributions (D) in the range of 2.1–3.4 [28,29]. This means that the polymer backbone of the CPAM was uniform.

The average diameter and polydispersity index of the polymers were obtained using dynamic light scattering (angle of 90° and ambient temperature of 25 °C). The particle size distribution of the CPAM-1,2,3 samples showed that the average diameter of the polymer particles was 239.4 nm, 193.4 nm, and 196.1 nm, respectively, and similar to other studies which had particle sizes of about 200–300 nm [4,30]. The particle size distribution was from 100 to 750 nm (Figure 6). The DLS results show that the CPAM particles sizes were uniform.

Figure 7 shows the SEM images of the CPAM samples. Figure 7A–C shows the SEM images of C3012, C4008, and C6008, respectively; Figure 7D–F shows the SEM images of CPAM-1 (DC = 21.85%), CPAM-2 (DC = 40.25%), and CPAM-3 (DC = 71.17%). The surface of the CPAM samples is similar to those of commercial products and CPAMs in other studies [12,14].

In addition, Figure 8 shows the nitrogen adsorption–desorption isotherms of the CPAM samples. The BET surface area results showed that the adsorption of C3012, C4008, C6008, CPAM-1, CPAM-2, and CPAM-3 were 0.4235 m^2^/g, 0.1740 m^2^/g, 0.5528 m^2^/g, 0.4861 m^2^/g, 0.8071 m^2^/g, and 1.3998 m^2^/g, respectively. The difference in BET surface area was due to different surface morphologies of the CPAMs.

Characterization by ^1^H−NMR, FTIR, SEM, BET, DLS, and GPC confirmed that CPAMs with the desired cationic degree and molecular weight were successfully synthesized.

### 3.2. Development of Response Surface Models for CPAM Optimization

In this study, 15 experiments were carried out with the input variables (monomer concentration, content of monomer cation, and content of initiator) and three outputs (cationic degree, molecular weight, and conversion) based on the Box–Behnken method [19,20,21] (Table 1).

ANOVA was used to assess how well the resulting model fit the data (Appendix A). Four thresholds were used to conclude statistically significance of a model: (1) *p*-value < 0.05 [31,32]; (2) Adequate Precision Value > 4.0 [33]; (3) R^2^ > 0.8; and (4) lack-of-fit value showing that the discreteness of data was not statistically significant.

A high R^2^ value, close to one, is preferred, which indicates that the quadratic model has been properly adjusted to the experimental data (Table 2). The model is normally considered reproducible if its coefficient of variance (CV) is less than 10%.

#### 3.2.1. Effects of Input Parameters on Cationic Degree

The influence of monomer, DMC, and AM concentrations on the cationic degree (DC) of the CPAM is depicted in Appendix A. It is evident that the monomer concentration had no effect on the DC value of the CPAM. However, significant effects were observed with variations in DMC and AM concentrations. The DC of the CPAM increased with increasing DMC concentration and decreased as the AM concentration increased.

#### 3.2.2. Effects of Input Parameters on M_w_ of Synthesized CPAM Emulsion

The molecular weight of the copolymer changed significantly with variations in monomer concentration and the content of DMC, as shown in Appendix A. When the monomer concentration increased from 15% to 25%, the molecular weight of the CPAM increased and reached a maximum within the range of 25% monomer concentration. However, when the monomer concentration was further increased to 35%, the molecular weight decreased. This can be explained as follows: at monomer concentrations below 25%, copolymer formation was initiated, leading to an increase in molecular weight. However, as the monomer concentration increased, the viscosity of the emulsion significantly increased, resulting in a higher collision frequency between monomer radicals and monomers that were unable to interact with the active chain group, leading to reduced chain propagation. At excessively high monomer concentrations, the heat generated during the reaction was not effectively dissipated to the external environment, causing superheating and subsequent reduction in the molecular weight of the CPAM.

The content of the cation monomer DMC directly influenced the molecular weight and cationic degree of the resulting product. The molecular weight of the CPAM decreased with an increase in cationic degree, attributed to the lower reactivity of the cation monomer compared to the acrylamide monomer. As the cationic monomer content increased, the positive charge density also increased, while the diffusion rate of the monomer to the growing chain decreased, resulting in a decrease in the chain growth rate and a subsequent reduction in the molecular weight of the CPAM.

With an increase in DMC content, the initiator content needed to be increased accordingly to obtain high-molecular-weight CPAM emulsions. Since the activity of the cation monomer is lower than that of the AM monomer and the chemical structure of the cation monomer is complex, the addition of an initiator is necessary to enhance the contact between the free radical and the growing chain, facilitating the production of high-molecular-weight CPAM emulsions.

#### 3.2.3. Effects of Input Parameters on Conversion

Appendix A shows that the conversion increased along with the concentration of the monomer. When the monomer concentration reached 25%, the conversion of the reaction reached the optimal value, and it tended to decrease sharply when the monomer concentration was increased to 35% (Appendix A). This can be explained as follows: as the concentration of the monomer increased, the chain growth and the number of consumed monomers increased, leading to an increase in the reaction yield. However, an excessive monomer concentration results in high emulsion viscosity, which inhibits vascular growth, thereby reducing the conversion efficiency of the reaction.

The conversion of the reaction was also affected by the concentration of the cation monomer—DMC. When the concentration of DMC reached 30–70%, the conversion of the reaction reached a high value (about 97%) (Appendix A and Table 1). However, when the DMC concentration increased to 90%, the conversion decreased to 80% because of the formation of copolymers with large molecular chains and the increase in solution viscosity, which impedes free radical movement. Additionally, the electrostatic repulsion effect and steric effect of the cation monomer hindered contact between monomers, resulting in reduced conversion.

### 3.3. Application of Response Surface Models

The results of the experiments demonstrated that the use of response surface models enables the identification of the optimal conditions for monomer, cation monomer, and initiator concentrations in order to synthesize the desired CPAM with a specific molecular weight, conversion, and cationic degree (Table 3). Compared to studies that use single-factor experiments to optimize the synthetic conditions for CPAMs [4,10,26,34], our study will save time and materials because our models reduce the number of experiments required to optimize the conditions for the desired CPAMs, while the single-factor approach requires more experiments to optimize each input parameter.

Previous research utilized response surface models to optimize the conditions for high-molecular-weight CPAMs, and high conversion rate and flocculation efficiency [7,10,12,14,35,36]. While molecular weight, conversion rate, and flocculation efficiency are crucial factors in CPAM production, the market demands CPAMs with varying cationic degrees for different purposes and price points. In order to fill this gap, our models determine the optimal conditions for synthesizing CPAM with low, medium, and high cationic degrees. Depending on the type of wastewater being treated, users can select a suitable CPAM with a specific cationic degree and apply the optimized synthesis conditions. This streamlined process minimizes both the time and costs associated with the production of the desired CPAM.

### 3.4. Testing and Evaluating the Ability of Synthesized CPAMs to Treat Wastewater 

The flocculation in the wastewater treatment process utilizing a CPAM emulsion is illustrated in Figure 9. The CPAM cationic flocculant operates through a charge neutralization mechanism, whereby the negative charges present in the colloidal particles are neutralized. As a result, the particles are encouraged to move closer to each other and form clusters. The long carbon backbone of CPAM, coupled with its high molecular weight (reaching several million Da), facilitates the formation of bridges through a chain bridging mechanism. These bridges play a crucial role in the creation of larger flocs, thereby enhancing the efficiency of the treatment process [37,38].

Figure 10 demonstrates that the synthesized CPAM samples exhibited comparable efficacy in treating wastewater when compared to commercial products (C3012, C4008, and C6008). CPAM-1 and C3012 were used for treating wastewater from industrial plants, CPAM-2 and C4008 for treating wastewater from the paper-making process, and CPAM-3 and C6008 for treating wastewater generated in the paint shop of a car manufacturing plant.

In Figure 10, at CPAM concentrations of 55–65 mg/L, the wastewater treatment efficiency reached the best performance with COD values of 22–30 mg/L, BOD values of 20–27 mg/L, TSS values of 19–30 mg/L, and turbidity values of 12–20 NTU. At this CPAM concentration range, the flocs formed by the CPAM molecules and colloid particles have sufficient mass to precipitate. When the concentration of CPAM was outside of 55–60 mg/L, the flocs formed by the CPAM molecules and colloid particles were small and suspended resulting in low performance of wastewater treatment (i.e., high values of COD, BOD, TSS, and turbidity).

The results of the wastewater treatment experiment indicate that the water quality after treatment, achieved by our CPAM products was similar to that of commercial CPAMs (C3012, C4008, and C6008), and the discharged water meets the target values [39,40,41].

## 4. Conclusions

The synthesis of desired CPAMs with different cationic degrees (21.58%, 40.25%, and 71.7%) was successfully achieved using Box–Behnken models. The optimal conditions for synthesis were determined as follows: monomer concentration of 25%, monomer cation content of 22.5%, 44.41%, and 77.61%, and initiator content of 0.475%, 0.48%, and 0.59%. The obtained CPAMs exhibited excellent effectiveness in treating industrial wastewater. These findings have significant implications in streamlining CPAM production, meeting market demands promptly, and reducing product costs.

## Figures and Tables

**Figure 1 polymers-15-02693-f001:**
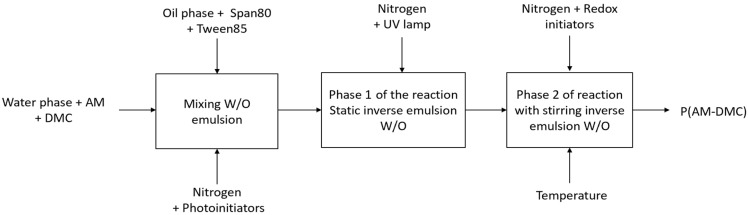
Workflow of synthesis process of the CPAM.

**Figure 2 polymers-15-02693-f002:**
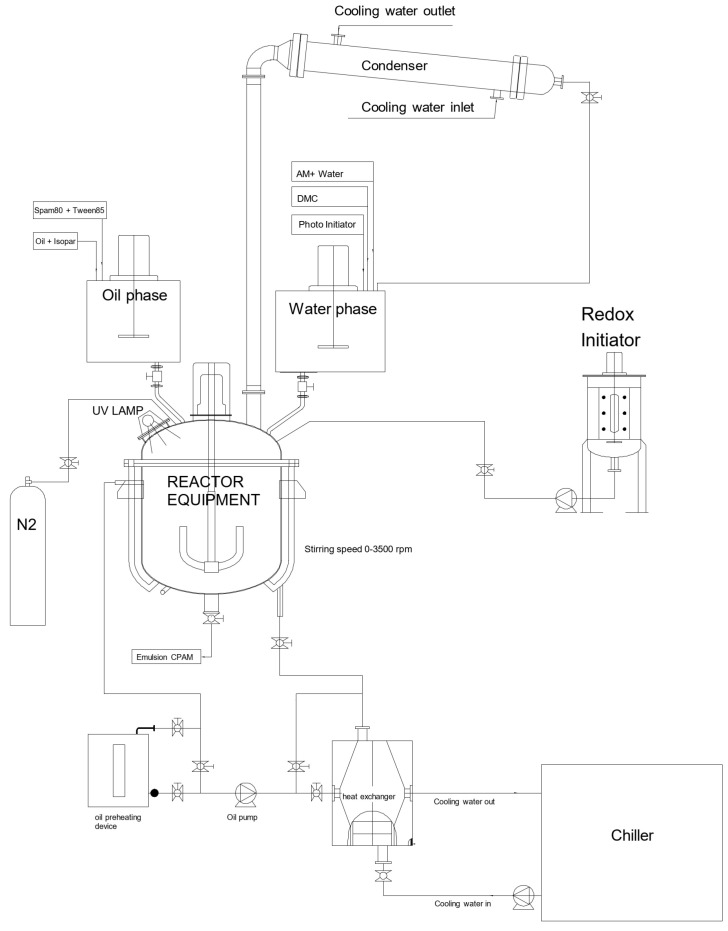
Device diagram for the CPAM synthesis process.

**Figure 3 polymers-15-02693-f003:**
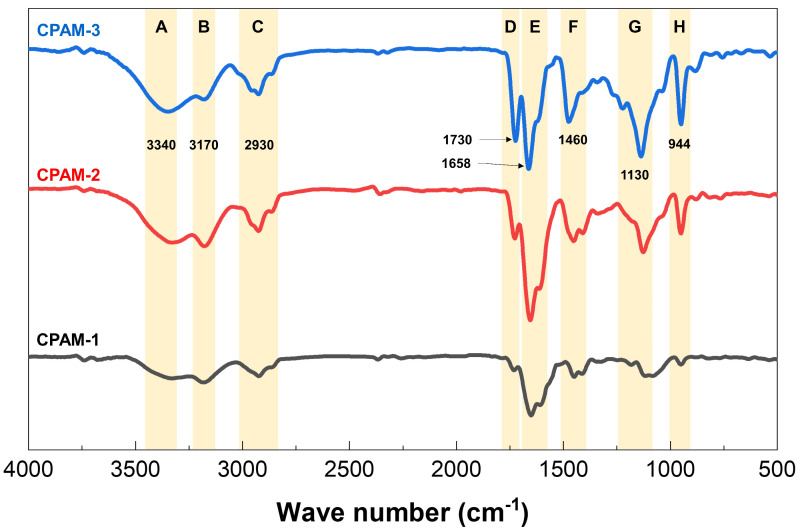
FTIR spectra of CPAM copolymers. Grey, red and blue spectra are data of CPAM-1,2,3, respectively. Eight identified peaks from 3500 to 900 cm^−1^ are assigned to peaks A–H.

**Figure 4 polymers-15-02693-f004:**
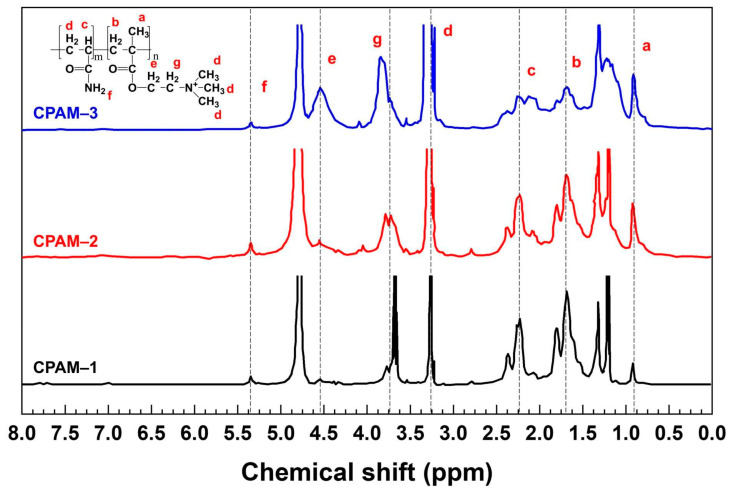
^1^H–NMR spectra of CPAM polymers with three different DC values.

**Figure 5 polymers-15-02693-f005:**
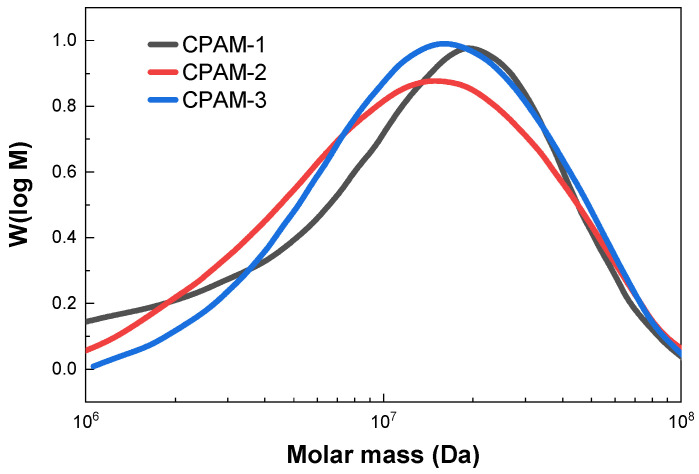
Molecular weight distribution of CPAM samples.

**Figure 6 polymers-15-02693-f006:**
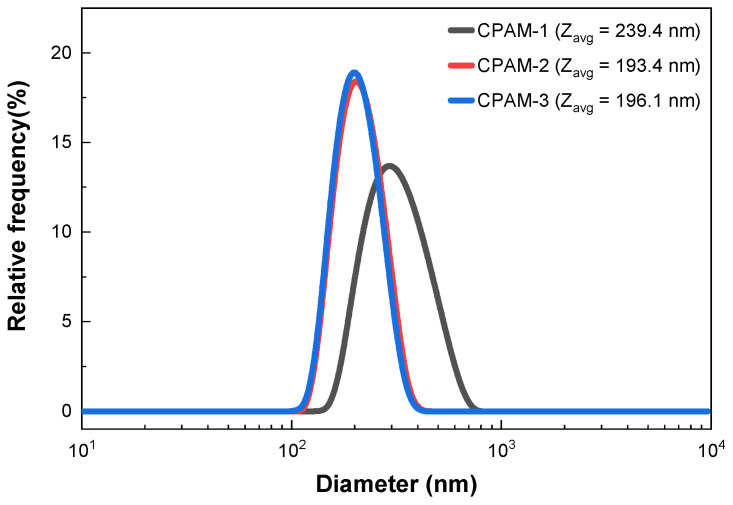
Dynamic light scattering data of CPAM samples.

**Figure 7 polymers-15-02693-f007:**
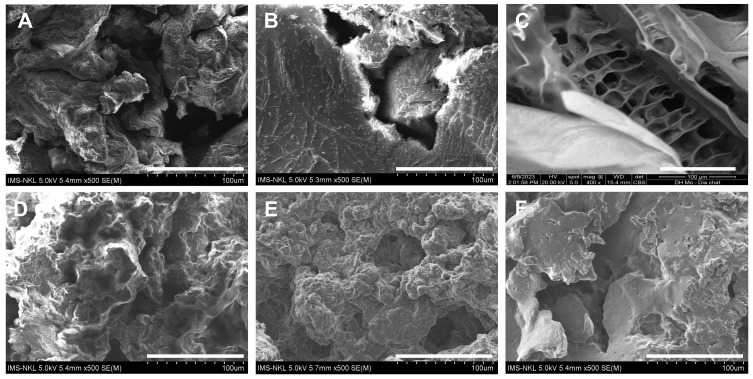
SEM images of CPAM samples: (**A**) C3012, (**B**) C4008, (**C**) C6008, (**D**) CPAM-1, (**E**) CPAM-2, and (**F**) CPAM-3. Scale bars are 100 µm.

**Figure 8 polymers-15-02693-f008:**
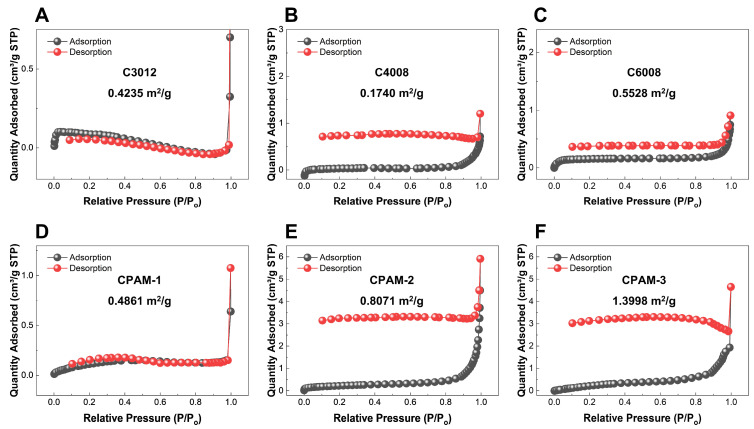
Nitrogen adsorption–desorption isotherms of three commercial CPAMs (C3012, C4008, and C6008) (**A**–**C**) and three synthesized CPAMs (CPAM-1,2,3) (**D**–**F**).

**Figure 9 polymers-15-02693-f009:**
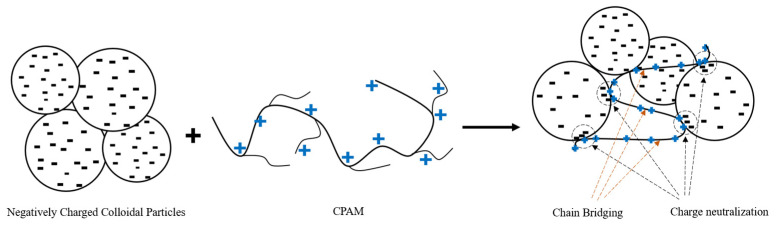
Mechanism of CPAM flocculation process. Circle objects are colloidal particles with negative charges (minus sign symbols), CPAM molecules show positive charges in molecular backbone (plus sign symbols).

**Figure 10 polymers-15-02693-f010:**
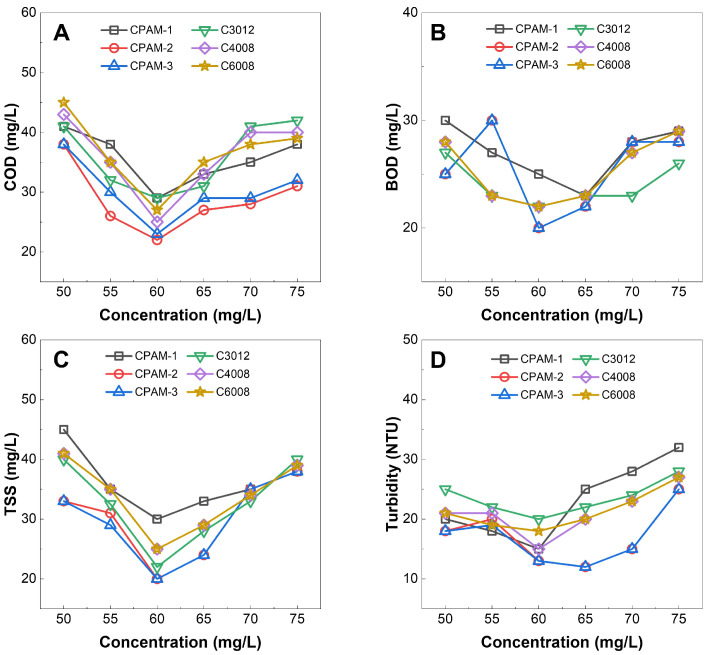
Wastewater treatment results using three synthesized CPAMs (CPAM-1,2,3) and three commercials CPAMs (C3012, C4008, and C6008). (**A**) COD index. (**B**) BOD index. (**C**) TSS index. (**D**) Turbidity.

**Table 1 polymers-15-02693-t001:** Results of the actual trial synthesis.

No	Monomer Concentration (%)	Content of Cationic Monomer (%)	Content of Initiator (%)	Cationic Degree (%)	Molecular Weight (Da)	Conversion (%)
1	25	90	0.45	78.38	3,908,260	77.36
2	15	10	0.55	7.98	3,014,500	84.27
3	25	10	0.45	8.51	9,010,940	90.5
4	35	10	0.55	7.78	5,563,790	77.84
5	25	50	0.55	47.55	11,007,000	97.01
6	25	90	0.65	83.68	8,008,260	91.05
7	35	90	0.55	81.64	4,003,350	80.45
8	15	50	0.65	40.89	4,006,450	85.67
9	35	50	0.45	43.15	6,300,900	80.43
10	25	50	0.55	49.79	10,769,520	96.51
11	35	50	0.65	43.1	6,500,900	81.72
12	25	50	0.55	48.02	10,687,000	96.09
13	25	10	0.65	6.13	4,209,400	74.28
14	15	90	0.55	74.9	3,000,570	83.77
15	15	50	0.45	44.23	4,876,450	89.08

**Table 2 polymers-15-02693-t002:** Goodness of fit of response surface models for the cationic degree, molecular weight, and conversion of CPAM (AP: adequate precision; CV: coefficient of variation).

Response	The Final Equation	R^2^	Adj. R^2^	Pred. R^2^	R^2^	Adj. R^2^
Cationic Degree	$\begin{array}{c} DC=-61.08+1.10A+0.65B+202.95C+\\ 4.34\times 10^{-3}AB+0.82AC+0.48BC-3.36\times \\ 10^{-2}A^{2}-1.26\times 10^{-3}B^{2}-225.54C^{2} \end{array}$	0.9984	0.9955	0.9779	50.5004	4.15
Molecular Weight	$\begin{array}{c} M_{w\;}=-4.70\times 10^{7}+1.94\;\times 10^{6}A-1.01\times \\ 10^{5}B+1.29\times 10^{8}C-9.67\times 10^{7}AB+\\ 2.68\times 10^{5}AC+5.56\times 10^{5}BC-3.90\times 10^{4}A^{2}\\ -1.89\times 10^{3}B^{2}-1.51\times 10^{7}C^{2} \end{array}$	0.9993	0.9980	0.9954	75.7478	2.02
Conversion	$\begin{array}{c} \%H_{\;}=-40.70+2.48A-0.56B+4.52\times 10^{2}C+\\ 1.93\times 10^{-3}AB+1.18\times AC+1.87BC-7.01\\ \times 10^{-2}A^{2}-4.96\times 10^{-3}B^{2}-5.30\times 10^{2}C^{2} \end{array}$	0.9977	0.9935	0.9706	45.5842	0.6899

**Table 3 polymers-15-02693-t003:** Experimental results of cationic degree, molecular weight, and conversion of CPAM.

				Experimental Results			Predicted Results		
DC (%)	B	A	C	DC (%)	MW (Da)	H (%)	DC (%)	MW (Da)	H (%)
Low	22.5	25	0.475	21.85	7,315,261	95.12	22.50	10,064,867	93.60
Medium	44.41	25	0.48	40.25	9,264,425	92.60	42.50	10,412,181	94.82
High	77.61	25	0.59	71.17	11,342,957	94.00	72.50	9,433,541	94.23

A: monomer concentration (%), B: content of cation monomer (%), and C: content of initiator (%).

## Data Availability

All data are contained within the article or Appendix A.

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
