# Peer review of "Optimization Conditions to Obtain Cationic Polyacrylamide Emulsion Copolymers with Desired Cationic Degree for Different Wastewater Treatments"

_polymers, 2023, doi:10.3390/polym15122693_

Round 1

Reviewer 1 Report

20230526-Comments to polymers:

Major revision is recommended to this submission as suggested by following items.

1. Removal mechanism for typical pollutant can be presented using Schematic figure.

2. The relationship between structure and application should be clearly established with supportive data.

3. SEM and nitrogen sorption analysis is needed to learn about the morphology and porosity of this kind of materials.

4. English writing can be further improved.

see

Author Response

Dear reviewers,

We appreciate valuable comments from the reviewers. Following yours comments, we have thoroughly revised our manuscript to improve its clarity. The modifications that we made were incorporated as red-colored texts in the revised manuscript. We also present our response to each comment of the reviewers as in the attached file.

Best regards,

Reviewer 2 Report

In this manuscript, the results of this research are conveyed thoughtfully and completely, and they are consistent with the experimental findings. However, the authors failed to explain and draw out the novelty of the work, this aspect needs to be improved. This work is worthwhile to be publish in this journal after minor revision. The following issues should be addressed:

1. Introduction is well-organized but the importance and novelty of the research should be highlighted and more clearly stated. The authors should give some examples of works in the bibliography, to clear the advantage of their work in comparison with those works.

2. Maybe the author should compare their results clearly with other reported works, highlighting the advantage and disadvantages of their novel composite.

3. The authors are responsible for the English, which should be polished throughout the manuscript to clear some minor typo/grammar errors.

Hence, I recommend it accepted for publication after minor revisions.

Author Response

(The authors gave the same response as above.)

Reviewer 3 Report

In this manuscript, the authors reported "Optimization conditions to obtain cationic polyacrylamide emulsion copolymer with desired cationic degree for different wastewater treatment". The work is interesting. I recommend it be accepted for publication after minor revision. The main concerns are as follows.

1.     Abstract: The authors should modify the abstract section to highlight the novelty of the research and its contribution accurately.

2.     Introduction: The introduction section needs to be rewritten to incorporate recent work and emphasize the significance of using cationic polyacrylamide. The authors should provide a comprehensive overview of the field and explain how this cationic polyacrylamide addresses current research gaps or challenges. 

3.     Results:

a.     In Figure 3: The authors did not note some of the peaks in FTIR.  

b.     In Figure 9: the DLS spectra of the prepared sample are not clear. Check it.

c.     The authors need to improve the figure quality of Figure 10.

4.     Discussion:

a.     The wastewater treatment studies should be discussed in more detail. The authors should provide a comprehensive analysis of the experimental results, discussing the mechanisms involved in the wastewater treatment process. Additionally, the authors should compare the performance of the cationic polyacrylamide with other relevant polymers, highlighting its superior or unique properties.

5.     General comments:

      a.   There are more typos in the manuscript; double-check it thoroughly.

b.     Improve the figure captions and add more details.

c.     The authors should polish the English carefully and thoroughly in the manuscript.

In summary, the manuscript requires modification in the abstract, introduction, and results sections to address the points mentioned above. By incorporating these suggestions, the manuscript will provide a clear understanding of the research and its implications in the field of cationic polyacrylamide.  

 The authors should polish the English carefully and thoroughly in the manuscript.

Author Response

(The authors gave the same response as above.)

Round 2

Reviewer 1 Report

The authors have addressed comments carefully and this revised version can be accepted for publication in the Journal.